# Spray-Dried Inclusion Complex of Apixaban with β-Cyclodextrin Derivatives: Characterization, Solubility, and Molecular Interaction Analysis

**DOI:** 10.3390/polym17212850

**Published:** 2025-10-26

**Authors:** Da Young Song, Jeong Gyun Lee, Kyeong Soo Kim

**Affiliations:** Department of Pharmaceutical Engineering, Gyeongsang National University, 33 Dongjin-ro, Jinju 52725, Republic of Korea; sdysny@naver.com (D.Y.S.); leepipi87@naver.com (J.G.L.)

**Keywords:** apixaban, β-cyclodextrin derivatives, spray drying, inclusion complex, solubility enhancement, molecular docking simulation

## Abstract

Apixaban (APX) is a direct oral anticoagulant with low aqueous solubility and limited bioavailability. This study aimed to improve APX solubility by forming spray-dried inclusion complexes (ICs) with β-cyclodextrin (β-CD) derivatives. ICs were prepared using hydroxypropyl-β-CD (HP-β-CD), sulfobutylether-β-CD (SBE-β-CD), randomly methylated-β-CD (RM-β-CD), and heptakis(2,6-di-O-methyl)-β-CD (DM-β-CD). Complex formation (1:1 stoichiometry) was confirmed by phase solubility studies and Job’s plots. The ICs were characterized by SEM, PXRD, DSC, and FTIR, and their saturated solubility was evaluated. Molecular docking assessed host–guest interactions. Among the tested carriers, DM-β-CD exhibited the highest stability constant (K_C_ = 371.92 M^−1^) and produced amorphous ICs. DM-ICs achieved the greatest solubility enhancement at all pH conditions, with a maximum solubility of 1968.7 μg/mL at pH 1.2 and ~78.7-fold increase in water compared with pure APX. Docking results supported stable inclusion with the lowest binding free energy (−8.01 kcal/mol). These findings indicate that DM-β-CD-based ICs effectively enhance APX dissolution and show potential as solubilizing carriers for oral dosage forms.

## 1. Introduction

Apixaban (APX) was approved by the U.S. Food and Drug Administration (FDA) in 2012 as a direct oral anticoagulant and is effective in reducing the risk of stroke and systemic embolism in patients with non-valvular atrial fibrillation [1]. APX exhibits poor aqueous solubility (28–29 µg/mL) and an oral bioavailability of approximately 50% (Figure 1a) [2,3,4,5]. Accordingly, strategies that enhance solubility, and thereby increase in vivo bioavailability, are essential to improve therapeutic performance and minimize interindividual variability in absorption. As the number of poorly water-soluble drug candidates continues to rise, identifying robust solubility-enhancement approaches has become increasingly important in pharmaceutical development. Among these, complexation with cyclodextrins (CDs) is a proven method for improving aqueous solubility and modulating physicochemical behavior [6,7].

CDs are cyclic oligosaccharides enzymatically produced from starch and composed of D-glucopyranose units linked by α-1,4-glycosidic bonds (Figure 1b) [8,9,10]. They adopt a truncated-cone (toroidal) architecture with a hydrophilic exterior and a relatively hydrophobic internal cavity, enabling non-covalent inclusion complex (IC) formation with hydrophobic guest molecules [11,12]. Through such inclusion, CDs can enhance the apparent aqueous solubility and alter both the solid-state and solution properties of poorly soluble drugs [12,13].

Although β-cyclodextrin (β-CD) is the most commonly employed member of this class, its low aqueous solubility imposes practical constraints on formulation [7,8]. Consequently, a range of β-CD derivatives incorporating tailored hydrophilic or hydrophobic substituents has been developed, many of which significantly increase the solubility of both the host and the resulting drug ICs (Figure 1c) [7,9].

Prior studies have examined APX solubilization with hydroxypropyl-β-CD (HP-β-CD); however, comprehensive head-to-head comparisons across distinct β-CD derivatives remain limited, and the capability of 2,6-di-O-methyl-β-CD (DM-β-CD) to solubilize APX has not been clearly defined. Motivated by evidence that methylated β-CDs can afford superior inclusion efficiency and solubilization for poorly soluble drugs, the present study conducted a controlled comparison of four β-CD derivatives—HP-β-CD, sulfobutylether-β-CD (SBE-β-CD), randomly methylated β-CD (RM-β-CD), and DM-β-CD—prepared under identical spray-drying conditions with the objective of identifying the optimal host for APX.

To quantify solubility enhancement and elucidate complex stoichiometry, phase-solubility studies and Job’s plot analyses were performed. Solid-state ICs were prepared by spray drying. Because APX is scarcely soluble in most organic solvents commonly used for spray drying, a ternary solvent system comprising dichloromethane (DCM), ethanol (EtOH), and water was employed. This approach follows Janssens et al. [14], who demonstrated that EtOH can act as a mutual cosolvent between immiscible organic and aqueous phases, enabling stable dispersions and homogeneous spray-drying feed solutions.

The spray-dried ICs were benchmarked against physical mixtures using SEM, PXRD, DSC, and FTIR to assess the morphological and solid-state transformations. In parallel, molecular docking simulations were conducted to interrogate the host–guest interactions between APX and each β-CD derivative. Following a derivative-specific grid-box definition strategy [15], docking parameters were tailored to the cavity dimensions of each β-CD to ensure adequate sampling of guest orientations within the host cavity. The calculated binding free energies were then correlated with the experimentally determined apparent stability constant (K_C_) and the solubility-enhancement outcomes of the spray-dried ICs, enabling a comprehensive comparison of β-CD derivatives as solubilizers for APX.

## 2. Materials and Methods

### 2.1. Materials

The study materials were sourced as follows: APX from Medichem (Barcelona, Spain); α-CD, β-CD, γ-CD, and HP-β-CD from Ashland Inc. (Wilmington, DE, USA); SBE-β-CD from GLPBIO Technology Inc. (Montclair, NJ, USA); and RM-β-CD, DM-β-CD, and TM-β-CD from TCI Chemicals Co. (Tokyo, Japan). Organic solvents—acetonitrile (ACN), ethanol (EtOH) and dichloromethane (DCM)—were supplied by Daejung Co. Ltd. (Siheung, Republic of Korea). Distilled water (D.W.) was produced using the laboratory distillation apparatus, and all other chemicals employed were of analytical grade.

### 2.2. HPLC Method for Sample Analysis

Quantitative analysis of APX was carried out on an HPLC system (Agilent 1260 series; Agilent Technologies, Santa Clara, CA, USA) equipped with a UV–Vis detector (Agilent G1314). Separation was achieved on a SunFire^®^ C18 column (4.6 × 150 mm, 5 µm; Waters Corporation, Milford, MA, USA). The mobile phase consisted of ACN and D.W (40:60, *v*/*v*). Flow rate, injection volume, and detection were 1.0 mL/min, 20 µL, and 280 nm, respectively [16]. Data were acquired and processed using OpenLab CDS ChemStation (C.01.08). The method was validated; the calibration curve was linear over 2.5–100 µg/mL with the regression equation y = 43.27x + 8.65 (R^2^ = 0.9999).

### 2.3. Phase Solubility Studies

The phase solubility experiment was conducted according to the method described by Higuchi and Connors to evaluate the effect of CDs on the solubility of APX [17]. An excess amount of APX was added to 1 mL of aqueous solutions containing various concentrations (0, 2.5, 5, 10, and 15 mM) of natural and derivative CDs including α-, β-, and γ-CDs as well as substituted β-CD derivatives. The mixtures were vortexed using a vortex mixer (Vortex-Genie 2; Scientific Industries, Inc., Bohemia, NY, USA) and subsequently shaken in a water bath (LSB-045S; DAIHAN LABTECH, Namyangju, Republic of Korea) at 37 °C and 75 rpm for 5 days. After equilibration, the samples were centrifuged at 13,500 rpm for 10 min to remove precipitates, and the clear supernatant was filtered through a 0.45 µm syringe filter (HYUNDAI MICRO, Anseong, Republic of Korea) to eliminate undissolved APX. All experiments were performed in triplicate. Prior to HPLC analysis, the filtrates were appropriately diluted with the mobile phase [18].

### 2.4. Job’s Plot (Continuous Variation Method)

The stoichiometric ratio of the APX–CD IC was determined using Job’s continuous variation approach [19,20]. Equimolar solutions of APX and CD (5 × 10^−5^ M) were combined in different molar ratios ranging from 0 to 1, while the total volume was kept constant at 1 mL. The absorbance of each mixture was measured at 280 nm, corresponding to the maximum absorption wavelength of APX, using a UV–vis spectrophotometer (UV-1800; Shimadzu, Kyoto, Japan). The Job’s plot was constructed by plotting ΔA·R against R, where ΔA denotes the absorbance difference between APX alone and the APX–CD mixture, and R is defined as [APX]/([APX] + [CD]) [21].

### 2.5. Inclusion Complex Formation

ICs of APX and CD were prepared using β-CD and its derivatives, based on the results of the phase solubility studies. The ICs were formulated at various molar ratios of APX to CD, ranging from 1:1 to 1:4. APX was dissolved in a DCM/EtOH 50/50 (*v*/*v*) solution, CD was dissolved in a D.W/EtOH 50/50 (*v*/*v*) solution, and the two solutions were mixed. The ICs were then manufactured using a Yamato ADL311SA spray dryer (Yamato Scientific Co. Ltd., Tokyo, Japan) via the spray drying method. For 1:1 ICs, the inlet temperature, outlet temperature, and atomizing air pressure were set to 80 °C, 60 °C, and 0.1 MPa, respectively. For 1:2, 1:3, and 1:4 ICs, the inlet temperature was adjusted to 85 °C. Detailed feed compositions for each formulation are provided in Table 1 [14].

### 2.6. Drug Entrapment Efficiency

The amount of APX incorporated into each IC was quantified to determine the drug content. A weighed portion of the IC, corresponding to 10 mg of APX, was dissolved in a mixture of ACN and D.W (40:60, *v*/*v*) and sonicated in an ultrasonic bath (POWERSONIC 520; HWASHIN INSTRUMENT Co. Ltd., Seoul, Republic of Korea) until a clear solution was obtained. The resulting solution was subsequently subjected to HPLC analysis under the chromatographic conditions described earlier [22,23]. The entrapment efficiency of APX within the ICs was then computed using Equation (1).(1)$$\mathrm{EE}\;(\%)=\frac{The\;amount\;of\;APX\;in\;IC}{actual\;APX\;amount\;uesd\;in\;formulation}$$

### 2.7. Scanning Electron Micrograph (SEM) Analysis

The surface morphology of pure APX, its physical mixture (PM, 1:1 *w*/*w*) with CD, and the ICs was investigated by SEM (FE-SEM, Tescan MIRA3; Kohoutovice, Czech Republic). For imaging, each specimen was affixed onto an aluminum stub using double-sided carbon adhesive tape and subsequently coated with a thin platinum film (6 nm/min for 4 min) under a vacuum pressure of 7 × 10^−3^ mbar with a sputter coater (K575X; EmiTech, Madrid, Spain) to improve surface conductivity.

### 2.8. Powder X-Ray Diffraction (PXRD) Analysis

The crystalline nature of APX, the PMs, and the ICs was analyzed by PXRD using an Ultima IV diffractometer (Rigaku, Tokyo, Japan) fitted with a Cu-Kα radiation source (λ = 1.54 Å). Measurements were performed at ambient temperature under operating parameters of 40 kV and 40 mA. Diffraction patterns were recorded across a 2θ range of 10–40° with a scanning speed of 0.02°/s.

### 2.9. Differential Scanning Calorimetry (DSC) Analysis

The thermal characteristics of each sample were examined using a differential scanning calorimeter (DSC Q20; TA Instruments, New Castle, DE, USA). Approximately 4–5 mg of sample was precisely weighed and hermetically sealed in an aluminum pan covered with a matching lid. The analysis was conducted from 50 °C to 300 °C at a heating rate of 10 °C/min under a continuous nitrogen flow of 20 mL/min.

### 2.10. Fourier Transform Infrared (FTIR) Analysis

FTIR spectroscopy was carried out to verify potential molecular interactions between APX and CD in the ICs. The spectra were recorded using a Spectrum Two™ spectrometer (PerkinElmer, Waltham, MA, USA) employing the potassium bromide (KBr) pellet technique. Around 1 mg of sample was blended with 200 mg of KBr and compressed into translucent pellets using a hydraulic press. The spectral data were collected within the wavenumber range of 4000–400 cm^−1^ at a resolution of 1 cm^−1^.

### 2.11. Saturated Solubility Study

The saturated solubility of APX, evaluated both as the free drug and as ICs, was assessed in pH 1.2, pH 4.0, pH 6.8, and D.W [24,25]. An excess amount of each IC corresponding to 10 mg of APX was transferred to a 2 mL Eppendorf tube, and 1 mL of the respective medium was added. The suspensions were vortexed and incubated at 37 °C in a shaking water bath (75 rpm) for 5 days. After equilibration, samples were centrifuged at 13,500 rpm for 10 min, and the supernatants were filtered through 0.45 µm syringe filters. Filtrates were appropriately diluted with the HPLC mobile phase and analyzed using the method described above.

### 2.12. Molecular Docking

The 3D structural data for APX were downloaded from the PubChem chemical database (https://pubchem.ncbi.nlm.nih.gov/) in SDF format and subsequently transformed to PDB format through Open Babel version 3.1.1. The resulting structure was energy-minimized by geometric optimization (steepest descent algorithm, 5000 steps) based on the Universal Force Field (UFF) in Avogadro version 1.2.0, and the optimization process was repeated until the total energy converged [26,27]. The 3D structure of β-CD was extracted from the protein complex (PDB ID: 1DMB) deposited in the Protein Data Bank (https://www.rcsb.org/) and separated using ChimeraX version 1.8 [28]. Derivative structures including HP-β-CD, SBE-β-CD, DM-β-CD, and RM-β-CD were reconstructed from the parent β-CD model and minimized under the same computational parameters as APX. Docking simulations between APX and each CD were performed using AutoDock 4 and AutoDock Vina implemented within AutoDockTools (MGLTools v1.5.7; Scripps Research Institute, La Jolla, CA, USA), which are commonly utilized to estimate ligand–receptor binding affinities [29,30]. In these simulations, APX was defined as a flexible ligand, whereas the CD molecules were treated as rigid hosts. Prior to docking, PDBQT input files were generated for both APX and CD [28]. During this step, nonpolar hydrogens were merged on CD atoms and Gasteiger–Marsili charges were assigned [31]. The APX molecule was characterized by defined atom types, hydrogen-bond donors and acceptors, aliphatic and aromatic carbons, and six rotatable bonds identified as N6–C20, C19–N5, C22–C28, C28–N9, N7–C27, and C33–O4 (Figure 2) [29].

To comprehensively explore potential conformations of APX inside the CD cavities, grid dimensions were set to 70 × 70 × 70 Å for HP-β-CD, 110 × 90 × 90 Å for SBE-β-CD, and 60 × 60 × 60 Å for both DM-β-CD and RM-β-CD, with a grid spacing of 0.375 Å. The grid box was configured to fully enclose the CD host and the entire APX/CD complex region [32]. To identify the most energetically favorable docking pose, the Lamarckian genetic algorithm (LGA) was executed with 100 iterations. The maximum number of energy evaluations and generations was set to 2.5 × 10^7^ and 27,000, respectively [33,34,35].

To justify the selected docking parameters, AutoDock 4.2 settings recommended by Morris et al. [33] were applied as the baseline (grid spacing = 0.375 Å, population size = 150, maximum generations = 27,000). The grid box was sized to fully enclose each β-CD derivative and the possible APX/CD complex region. According to previous reports, apixaban is a non-ionized/non-ionizable compound [36,37]. Therefore, pH-dependent ionization or conformational changes were not considered. The complex exhibiting the minimum binding free energy among the docking outcomes was selected and visualized using ChimeraX version 1.8 [38,39].

## 3. Results

### 3.1. Drug–Cyclodextrin Interaction via Phase Solubility and Job’s Plot Studies

The phase solubility profiles describing the complexation between APX and different CDs were interpreted on the basis of the classification proposed by Higuchi and Connors (Figure 3). In this system, Type A curves denote the generation of water-soluble inclusion complexes, which can be subdivided into three forms depending on the solubility trend.

An A_L_-type pattern is characterized by a linear enhancement in drug solubility as the CD concentration increases, while upward (A_P_-type) or downward (A_N_-type) curvatures reflect positive or negative deviations from linearity, respectively [40,41,42].

An enhancement in APX solubility was observed with higher concentrations of α-, β-, and γ-CD, as presented in Figure 4a. Among them, β-CD exhibited the most pronounced solubility enhancement, leading to further studies using its derivatives. The comparative solubility ranking of β-CD derivatives for APX was determined as DM-β-CD > RM-β-CD > HP-β-CD > SBE-β-CD > TM-β-CD, with respective values of 160.0 ± 0.7, 152.6 ± 3.6, 116.3 ± 1.1, 108.6 ± 0.3, and 41.7 ± 0.2 µg/mL. As a result, all β-CD derivatives exhibited A_L_-type phase solubility profiles, in which the solubility of APX increased linearly with CD concentration (Figure 4b).

This profile refers to the formation of a 1:1 IC between the drug and CD, and this complex formation can be expressed by Equations (2) and (3) [43]:[D] + [CD] ⇄ [D:CD](2)(3)$$K_{C}=\frac{\left[D\text{:}CD\right]}{\left[D\right][CD]}$$

[D:CD] represents the complex formed between the drug and CD. [D], [CD], and [D:CD] refer to their respective equilibrium concentrations, and Kc denotes the apparent stability constant for complex formation. The Kc was calculated from the slope of the linear region of the phase solubility plot according to Equation (4) [44,45,46], where S_0_ (0.052 mM) represents the intrinsic solubility of APX without CD.(4)$$K_{C}=\frac{Slope}{S_{0}(1-Slope)}$$

The Kc values were mostly in the range of 50–2000 M^−1^ [47], and the calculated results for DM-β-CD (371.92 M^−1^), RM-β-CD (357.35 M^−1^), HP-β-CD (254.73 M^−1^), SBE-β-CD (232.94 M^−1^), and TM-β-CD (49.29 M^−1^) fell within this generally reported range. A low Kc value indicates weak interactions between the CD and the drug, suggesting the presence of a relatively high amount of free, non-included drug. In contrast, a high Kc value implies that complex formation is predominant [48].

The Gibbs free energy change (ΔG°) represents the change in free energy associated with the transfer of the drug into the CD cavity in an aqueous solution and serves as an indicator of the spontaneity of IC formation [49,50]. ΔG° was calculated from Kc using Equation (5):ΔG° = −RT ln Kc(5)

Equation (5) includes R, the gas constant (8.314 J/mol·K), and T, the temperature used in the experiment (310.15 K, equivalent to 37 °C). When ΔG° is negative, the complex forms spontaneously, indicating an exothermic process mainly associated with hydrogen-bonding and hydrophobic interactions [51]. The slope, Kc, R^2^ value, and calculated ΔG° for each cyclodextrin were obtained from the phase solubility diagrams and are summarized in Table 2.

TM-β-CD exhibited the lowest K_C_ value, indicating the weakest interaction with the drug and suggesting that a larger proportion of the drug remains in the free, uncomplexed form. The glucopyranose units of β-CD adopt a ^4^*C*_1_ chair conformation, in which all glucose units are arranged in the same orientation, allowing the O2 and O3 hydroxyl groups to be positioned on the same face along the glycosidic linkage. This facilitates the formation of intramolecular O2(n)…O3(n−1) hydrogen bonds that contribute to the structural stability of the macrocycle. In contrast, TM-β-CD is a derivative in which the O2, O3, and O6 hydroxyl groups of the glucose residues are all methylated. When the hydroxyl groups at O2, O3, and O6 are all methylated, the formation of such hydrogen bonds becomes impossible. As a result, the regular circular structure of β-CD collapses, leading to an asymmetric, distorted elliptical shape with a significantly reduced cavity volume [52,53,54,55]. These structural characteristics suggest that TM-β-CD is unsuitable for the formation of stable IC, which may explain the observations in the phase solubility study. Therefore, it was excluded from the preparation of inclusion complexes.

Using Job’s plot, the stoichiometric composition of the APX–CD complex was assessed. The molar fraction (R) at which the curve reaches its maximum represents the stoichiometric ratio of the complex. The maximum absorbance peak was observed at R = 0.5 (Figure 5), indicating that an IC was formed at a 1:1 ratio between APX and CD [56]. This result was consistent with the stoichiometry of the complex determined from the phase solubility study [57].

### 3.2. Evaluation of Drug Entrapment Efficiency

The entrapment efficiency (EE) of the IC is presented in Table 3. The EE values ranged from 93.0 ± 0.2% to 100.3 ± 1.5%. These results indicate that all prepared formulations exhibited high drug incorporation efficiency. The narrow standard deviation values suggest uniform drug distribution within the complexes and consistent formulation performance across different β-CD derivatives.

### 3.3. SEM Analysis

SEM images of the APX, β-CD derivatives, physical mixtures (PMs), and ICs were obtained to observe the morphological differences among the samples (Figure 6). APX exhibited small, irregular crystalline particles less than approximately 5 µm in size, while β-CD derivatives appeared as large, amorphous particles with nearly spherical shapes. The PMs showed a surface morphology in which small APX drug particles were scattered on the surface of the CD particles, indicating the absence of significant inclusion interactions between APX and CD. In contrast to the PMs, the ICs exhibited a distinct morphological difference. The ICs prepared via spray drying were characterized by small, homogeneous, amorphous particles [58,59], and the original forms of the individual components were no longer distinguishable. Representative SEM images at ×10,000 showed that DM-IC and RM-IC formed relatively uniform spheres (about 2.5–5 µm), HP-IC was comparatively uniform, while SBE-IC exhibited a broad, polydisperse size distribution (about 1 µm to over 5 µm). These results indicate the formation of ICs and support the presence of inclusion interactions between the APX and β-CD derivatives [60].

### 3.4. PXRD Analysis

PXRD is an effective technique for confirming complexation using powdered samples and provides evidence for the formation of ICs. APX exhibited sharp characteristic peaks at 12.8°, 13.9°, 16.9°, 18.4°, 21.1°, 21.5°, 22.1°, 24.7°, and 26.9°, indicating that the drug was in a crystalline state (Figure 7). In contrast, each β-CD derivative showed no distinct diffraction peaks, suggesting an amorphous state. In the case of PMs, the diffraction patterns appeared as a superposition of those of APX and CD. The reduced intensity, slight shift, or disappearance of APX peaks may be attributed to the dilution effect of CD during sample preparation [61]. However, the PXRD patterns of the APX–CD ICs obtained via spray drying showed that all of the characteristic peaks of APX disappeared, and amorphous patterns similar to those of the respective β-CD derivatives were observed. The diffraction pattern of a complexed system is distinguishable from a simple overlay of each component’s individual patterns [62]. These results suggest that ICs may have formed as a result of interactions between the drug and CD, which is consistent with findings reported in other studies [63,64].

### 3.5. DSC Analysis

The DSC curve of APX exhibited an endothermic peak near its melting temperature (approximately 238 °C), which is characteristic of its crystalline nature [65]. A wide endothermic band appearing between 60 and 120 °C was observed for all β-CD derivatives, corresponding to the release of water molecules trapped within the internal cavity of CD [66,67,68,69]. SBE-β-CD, APX/SBE-β-CD PM, and SBE-ICs exhibited a peak corresponding to the thermal decomposition of sodium salts at approximately 269 °C (Figure 8b) [70]. The DSC curve of the APX/SBE-β-CD PM resembled the superimposed thermal profiles of APX and CD, suggesting no significant interaction between the two components [71,72]. The DSC curves of the HP-β-CD, DM-β-CD, and APX/RM-β-CD PMs showed that the endothermic peak of APX (238 °C) shifted to 235 °C for APX/HP-β-CD PM (Figure 8a), 230 °C for APX/DM-β-CD PM (Figure 8c), and 231 °C for APX/RM-β-CD PM (Figure 8d), with the peak range also broadening. These results suggest that there is weak interaction within the PMs [73].

In all ICs, the DSC curves showed that the distinct endothermic peak of APX completely disappeared. The complete disappearance of the melting peak of the crystalline drug in the DSC curves of ICs was interpreted as evidence that the drug molecule has been inserted into the CD cavity [74,75]. This supports the results of the PXRD study [76].

### 3.6. FTIR Spectroscopy Analysis

FTIR spectroscopy can confirm the IC formation by detecting vibrational changes that indirectly indicate drug–CD interactions [74]. Figure 9 presents the FTIR spectra of the APX, β-CD derivatives, PMs, and ICs. The FTIR spectrum for APX showed asymmetric and symmetric stretching vibrations of primary amines (N–H) at 3484 cm^−1^ and 3312 cm^−1^, respectively [77]. Peaks observed at 2910 cm^−1^ and 2867 cm^−1^ represent the asymmetric and symmetric C–H stretching of methyl (CH_3_) groups. The absorption bands within 1518–1398 cm^−1^ are attributed to the C=C stretching of the benzene ring, and those around 1257 cm^−1^ and 1682 cm^−1^ correspond to N–C and C=O stretching of the amide group [77,78].

The FTIR spectra for the PMs showed the characteristic peaks of both APX and the β-CD derivatives, indicating that simple mixing does not result in significant molecular interactions [79]. In contrast, noticeable changes were observed in the characteristic peaks of APX in the spectra of the ICs. The 3484 cm^−1^ and 3312 cm^−1^ bands (N–H asymmetric and symmetric stretching vibrations) disappeared due to overlap with the strong and broad O–H stretching vibration band of the β-CD derivatives around 3400 cm^−1^, suggesting the possibility of hydrogen bond formation between the two molecules [80]. Additionally, the 2910 cm^−1^ and 2867 cm^−1^ bands (C–H stretching vibrations) were reduced due to overlap with the C–H vibrations of β-CD derivatives near 2930 cm^−1^. The 1257 cm^−1^ and 1682 cm^−1^ bands as well as those between 1518 and 1398 cm^−1^ also showed decreased peak intensities.

These changes in the IR spectra are considered to result from the restricted vibrational freedom of the drug molecules upon inclusion within the CD cavity, which can be attributed to the disruption of the original intramolecular hydrogen bonding in APX and the subsequent formation of novel interactions upon inclusion complexation. These findings indicate that particular functional groups of APX are included within the CD cavity, leading to the formation of an IC [81,82]. Similar spectral changes have been observed in the CD ICs of various drugs [83,84,85], supporting the formation of ICs.

### 3.7. Saturated Solubility Study Results

The saturated solubility of APX and its ICs with various β-CD derivatives was evaluated in D.W and pH 1.2, 4.0, and 6.8 solutions. Under all examined pH conditions, pure APX maintained very low solubility, with measured concentrations below roughly 31 μg/mL, indicating that the drug was scarcely soluble across the tested range of pH values (Figure 10). Specifically, the solubility of APX was 24.0 μg/mL in D.W, 27.0 μg/mL at pH 1.2, 30.9 μg/mL at pH 4.0, and 24.9 μg/mL at pH 6.8.

In the case of the inclusion complexes, their solubility increased as the molar ratio of CD relative to the drug increased from 1:1 to 1:4. Among the β-CD derivatives, ICs prepared with DM-β-CD exhibited the highest solubility at all pH conditions, which is thought to be due to the relatively high affinity of these derivatives with APX, enabling more effective inclusion formation. Specifically, the solubility of DM-IC (1:4 molar ratio) was 1892.0 μg/mL in D.W, 1968.7 μg/mL at pH 1.2, 1912.2 μg/mL at pH 4.0, and 1895.8 μg/mL at pH 6.8, corresponding to approximately 78.7-, 72.9-, 62.0-, and 76.0-fold increases, respectively, compared with pure APX. On the other hand, the complexes prepared with HP-β-CD and SBE-β-CD exhibited a comparatively lower solubility enhancement than those with DM-β-CD and RM-β-CD. This was particularly evident in the APX/SBE-β-CD complex, where the increase in solubility was limited even at a 1:4 molar ratio. In addition, most complexes showed little variation in solubility with pH changes, exhibiting similar solubility increase trends across the entire range from pH 1.2 to 6.8. This indicates that the solubilizing effect of the ICs is largely unaffected by changes in pH conditions. Such characteristics imply that the prepared ICs may enhance drug solubility throughout the wide pH range of the gastrointestinal tract, thereby demonstrating their suitability for oral dosage form [86,87].

The saturated solubility results showed DM-β-CD to be the most effective β-CD derivative to improve the solubility of APX, a finding consistent with the phase solubility studies.

### 3.8. Molecular Docking Results

Molecular docking simulations were performed to investigate the molecular-level interactions involved in IC formation and to determine the orientation of APX (guest) within the cavity of various types of CD (host). The ICs of APX with four β-CD derivatives were subjected to docking analysis through the AutoDock software. To evaluate the binding affinity of each IC, calculations were performed for several energy terms including intermolecular energies (van der Waals energy and hydrogen bonding), solvation-related energies (desolvation), electrostatic energy, torsional energy, and unbound system energy. These values were used to compare and analyze the estimated free energy of binding and the final intermolecular energy. The docked structures are shown from side and bottom views in Figure 11, and the estimated free binding energy and final intermolecular energy for each complex are summarized in Table 4.

In the HP-IC, APX was accommodated within the CD cavity, and the amide group’s polar hydrogen participated in hydrogen bonding (2.763 Å) with the hydroxyl moiety of HP-β-CD. The aromatic rings, phenyl ring, and methoxyphenyl ring were located inside the cavity where hydrophobic interactions are possible. For the SBE-IC, APX was inserted horizontally into the cavity, forming a single hydrogen bond at a distance of 1.990 Å. Although this hydrogen bond was shorter than that observed in the HP-β-CD complex, the relatively higher docking energy suggested a lower overall binding stability. The RM-IC displayed two hydrogen bonds (2.187 Å and 2.107 Å) linking the amide functionality of APX with the hydroxyl groups located on the CD structure. The DM-IC exhibited one hydrogen bond (2.154 Å), while the triazole, phenyl, and pyridone rings of APX were positioned deep inside the hydrophobic cavity of DM-β-CD. The amide group was oriented toward the wider edge, forming a hydrogen bond with the CD (Figure 11d). This binding pattern supports the formation of a stable IC. Among the β-CD derivatives tested, DM-β-CD showed the lowest estimated free binding energy (−8.01 kcal/mol). This result aligned with its highest apparent stability constant (Kc = 371.92 M^−1^) and the greatest solubility increase (~78.7-fold). This agreement between the computational and experimental results supports that the stronger host–guest interaction of DM-β-CD contributes to its superior solubility enhancement [88].

## 4. Conclusions

In this study, spray-dried ICs of APX with four β-CD derivatives (HP-β-CD, SBE-β-CD, RM-β-CD, and DM-β-CD) were prepared, and the formation of the ICs and their solubility-enhancing effects on APX were evaluated. The phase solubility study revealed that every β-CD derivative followed an AL-type profile, suggesting a 1:1 complexation with APX. This finding was further validated by the Job’s plot analysis. Notably, DM-β-CD exhibited the highest apparent stability constant (K_C_ = 371.92 M^−1^), demonstrating superior inclusion efficiency. Solid-state characterization (SEM, PXRD, DSC, and FTIR) confirmed successful complex formation and distinct structural transformations compared with the PM. In the saturated solubility studies, DM-β-CD showed the highest solubility among all β-CD derivatives under all tested pH conditions, with the greatest increase of approximately 78.7-fold in D.W. Molecular docking analysis revealed that the DM-IC exhibited the lowest estimated binding free energy (−8.01 kcal/mol), indicating the most stable structure, consistent with the experimental solubility results. The docking simulations also visualized the host–guest orientation and confirmed the inclusion of APX within the β-CD cavity, with hydrogen bonding between the APX amide group and CD hydroxyl groups together with extensive hydrophobic contacts contributing to complex stabilization.

In conclusion, DM-β-CD was identified as the most effective β-CD derivative for forming stable ICs with APX, exhibiting the strongest host–guest interactions and the greatest solubility enhancement. These findings suggest that DM-β-CD-based ICs could serve as promising solubilizers for oral formulations of poorly soluble drugs. However, further dissolution and pharmacokinetic evaluations are required to validate their in vivo performance.

## Figures and Tables

**Figure 1 polymers-17-02850-f001:**
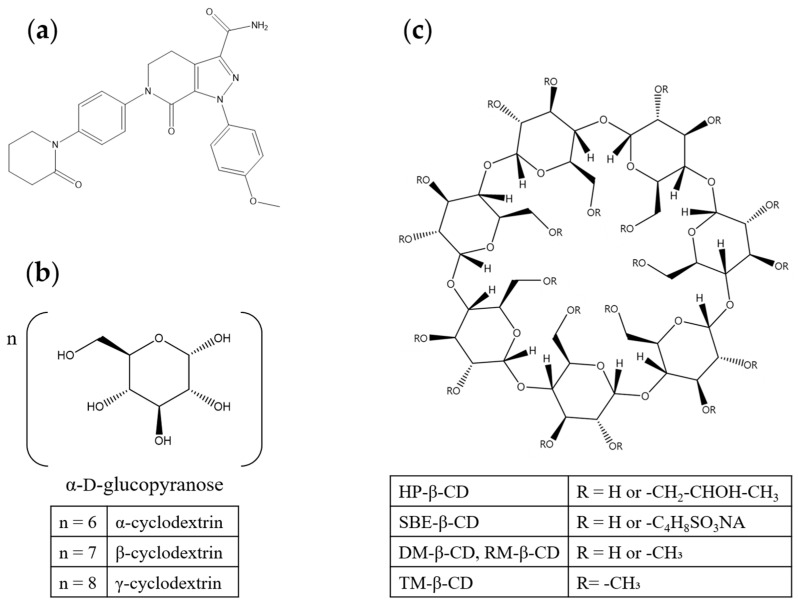
Chemical structures of (**a**) apixaban, (**b**) natural cyclodextrins, and (**c**) β-CD derivatives.

**Figure 2 polymers-17-02850-f002:**
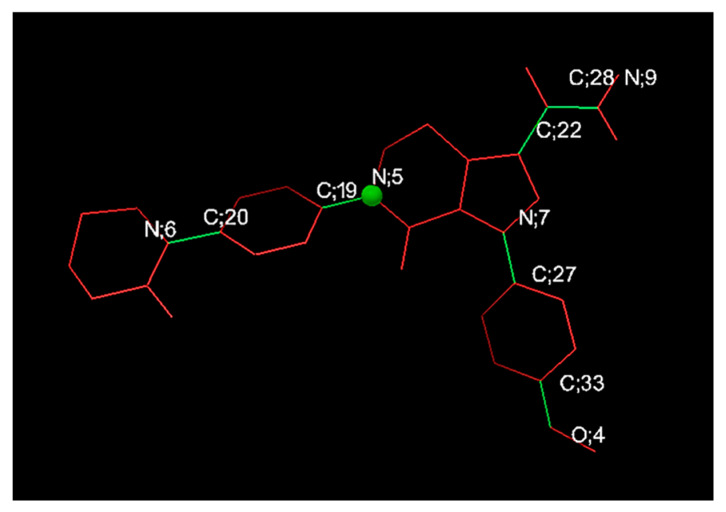
Rotatable bonds of apixaban (Green bonds represent rotatable bonds, while red bonds are non-rotatable).

**Figure 3 polymers-17-02850-f003:**
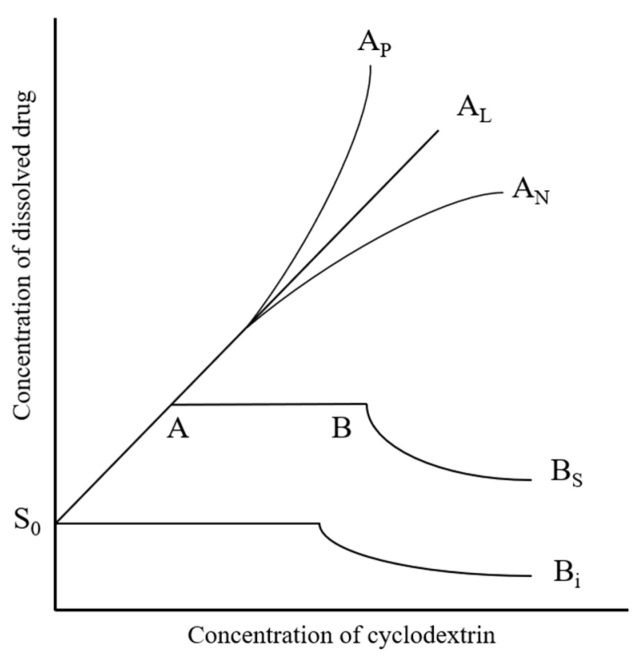
Classification of phase solubility profiles based on the Higuchi–Connors model.

**Figure 4 polymers-17-02850-f004:**
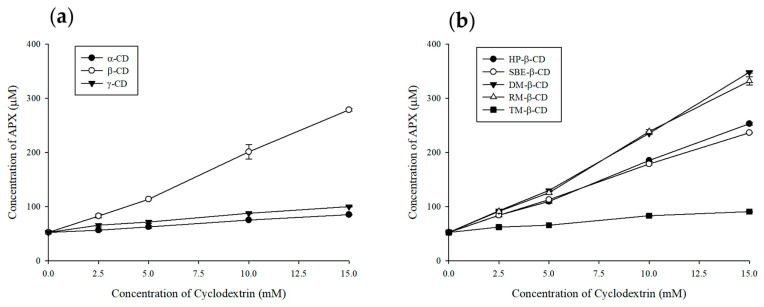
Phase solubility diagram of APX at different concentrations of cyclodextrin. Results are presented as the mean ± standard deviation (*n* = 3). (**a**) natural cyclodextrins and (**b**) β-CD derivatives.

**Figure 5 polymers-17-02850-f005:**
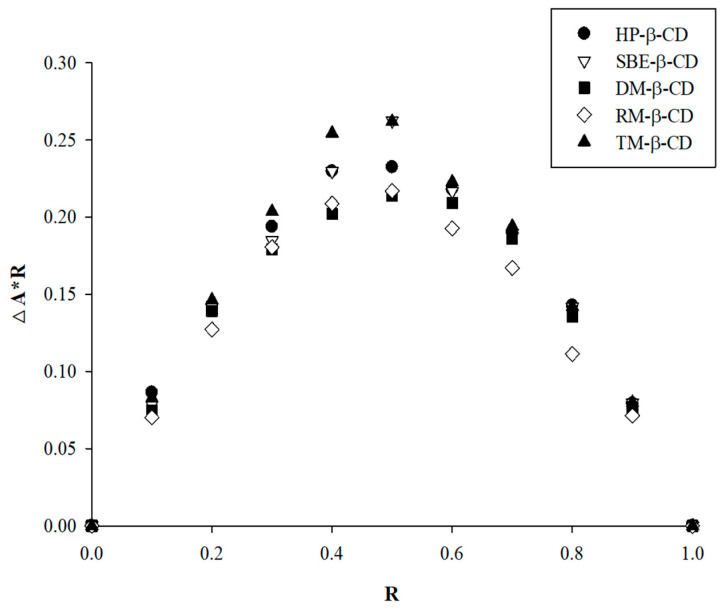
Job’s plot for the stoichiometric analysis of the inclusion complex formation between the APX and β-CD derivatives.

**Figure 6 polymers-17-02850-f006:**
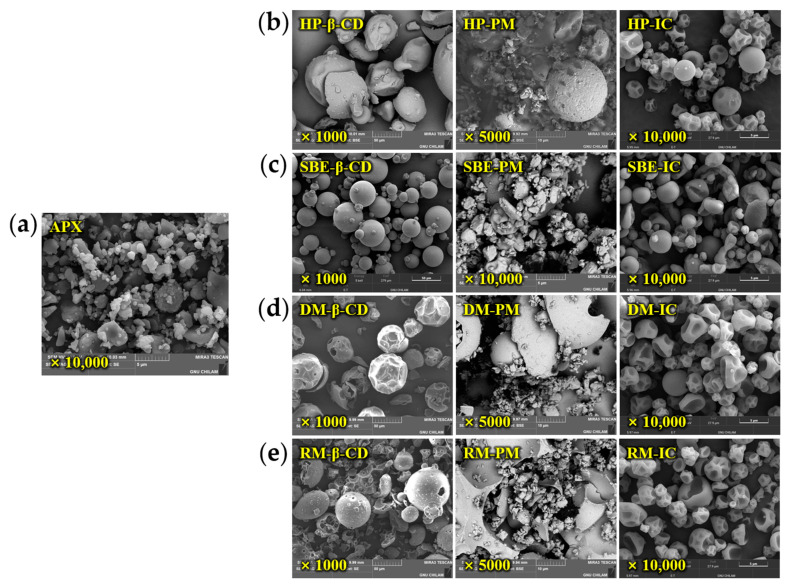
SEM images of APX (×10,000), β-CD derivatives (×1000), PMs (physical mixtures, ×5000; SBE-PM at ×10,000), and ICs (inclusion complexes, ×10,000): (**a**) APX; (**b**) HP-β-CD, HP-PM, and HP-IC; (**c**) SBE-β-CD, SBE-PM, and SBE-IC; (**d**) DM-β-CD, DM-PM, and DM-IC; (**e**) RM-β-CD, RM-PM, and RM-IC.

**Figure 7 polymers-17-02850-f007:**
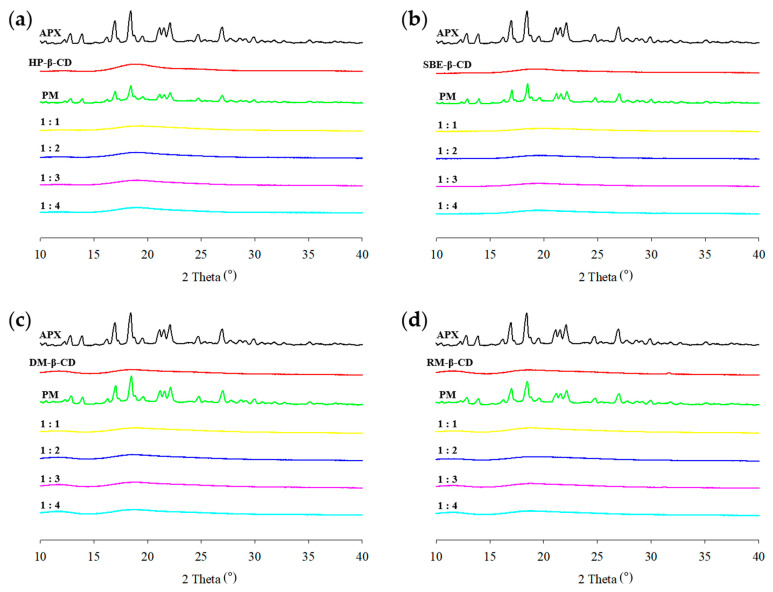
PXRD patterns of APX, β-CD derivatives, PMs (physical mixtures), and ICs (inclusion complexes): (**a**) HP-ICs (APX:HP = 1:*x*); (**b**) SBE-ICs (APX:SBE = 1:*x*); (**c**) DM-ICs (APX:DM = 1:*x*); (**d**) RM-ICs (APX:RM = 1:*x*).

**Figure 8 polymers-17-02850-f008:**
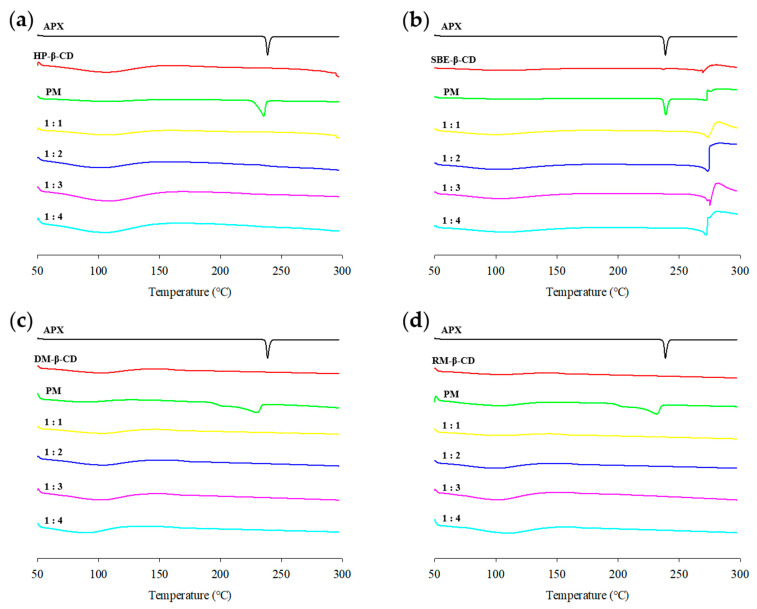
DSC curves of APX, β-CD derivatives, PMs (physical mixtures), and ICs (inclusion complexes): (**a**) HP-ICs (APX:HP = 1:*x*); (**b**) SBE-ICs (APX:SBE = 1:*x*); (**c**) DM-ICs (APX:DM = 1:*x*); (**d**) RM-ICs (APX:RM = 1:*x*).

**Figure 9 polymers-17-02850-f009:**
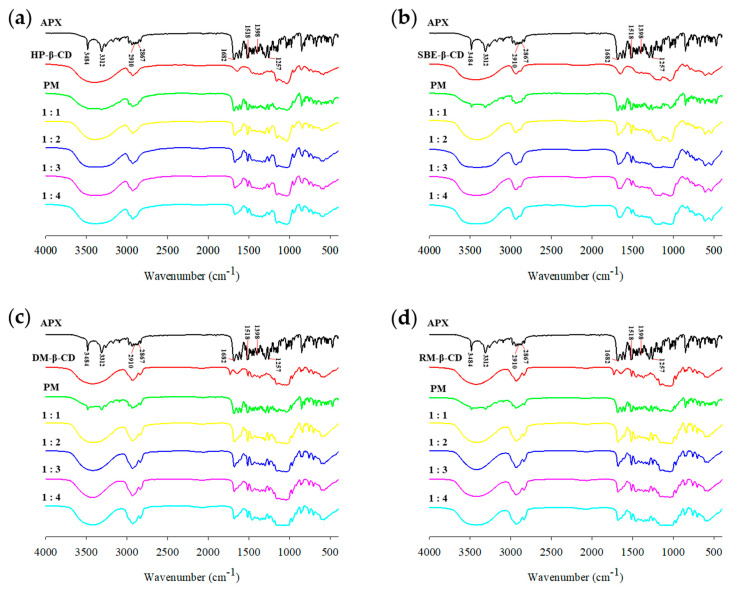
FTIR spectra of APX, β-CD derivatives, PMs (physical mixtures), and ICs (inclusion complexes): (**a**) HP-ICs (APX:HP = 1:*x*); (**b**) SBE-ICs (APX:SBE = 1:*x*); (**c**) DM-ICs (APX:DM = 1:*x*); (**d**) RM-ICs (APX:RM = 1:*x*).

**Figure 10 polymers-17-02850-f010:**
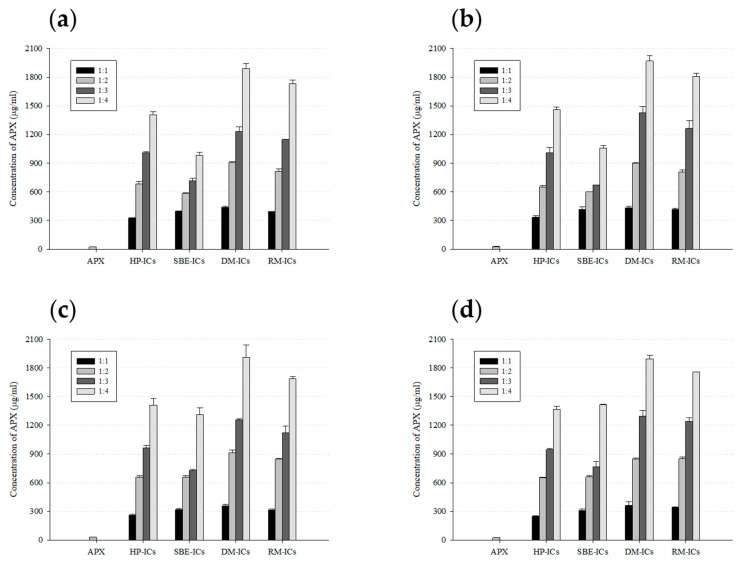
Solubility profiles of APX and ICs (inclusion complexes) in D.W and buffer solutions with pH values of 1.2, 4.0, and 6.8. Results are presented as the mean ± standard deviation (*n* = 3). (**a**) D.W, (**b**) pH 1.2, (**c**) pH 4.0, and (**d**) pH 6.8.

**Figure 11 polymers-17-02850-f011:**
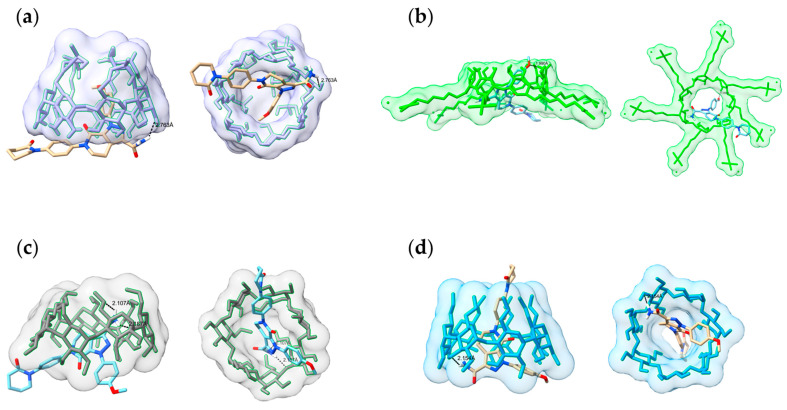
Molecular docking of the lowest binding energy conformation diagram of APX with β-CD derivatives (side and bottom views): (**a**) HP-IC; (**b**) SBE-IC; (**c**) DM-IC; (**d**) RM-IC.

**Table 1 polymers-17-02850-t001:** Formulation composition of the APX–β-cyclodextrin derivative inclusion complexes (“*x*” represents the molar ratio of CD).

Materials (mg)	HP (1:*x*)	SBE (1:*x*)	DM (1:*x*)	RM (1:*x*)
APX	2.5	2.5	2.5	2.5
HP-β-CD	7.5·*x*	-	-	-
SBE-β-CD	-	11.8·*x*	-	-
DM-β-CD	-	-	7.2·*x*	-
RM-β-CD	-	-	-	7.1·*x*
DCM:EtOH (1:1)	(150)	(200)	(150)	(150)
D.W:EtOH (1:1)	(150)·*x*	(200)·*x*	(150)·*x*	(150)·*x*
Total	2.5 + 7.5·*x*	2.5 + 11.8·*x*	2.5 + 7.2·*x*	2.5 + 7.1·*x*

**Table 2 polymers-17-02850-t002:** Apparent stability constant (K_C_), slope, coefficient of determination (R^2^), and Gibbs free energy change (ΔG°) values for each inclusion complex.

Cyclodextrin	^1^ Slope	K_C_ (M^−1^)	R^2^	ΔG° (kJ/mol)
HP-β-CD	0.0135	254.73	0.9976	−14.29
SBE-β-CD	0.0123	232.94	0.9995	−14.06
DM-β-CD	0.0199	371.92	0.9942	−15.26
RM-β-CD	0.0191	357.35	0.9947	−15.16
TM-β-CD	0.0026	49.29	0.9768	−10.05

^1^ Slope values are dimensionless, calculated as the ratio of solubility increase (µM) to cyclodextrin concentration (mM).

**Table 3 polymers-17-02850-t003:** Entrapment efficiency (%) of the APX/CD inclusion complexes at different molar ratios. Results are presented as the mean ± standard deviation (*n* = 3).

	**1:1**	**1:2**	**1:3**	**1:4**
HP-β-CD	96.5 ± 1.6	93.0 ± 0.2	94.9 ± 2.0	95.7 ± 1.2
SBE-β-CD	98.5 ± 1.3	93.6 ± 0.1	97.8 ± 0.4	97.0 ± 0.5
DM-β-CD	97.4 ± 0.7	94.8 ± 0.8	98.4 ± 1.7	100.3 ± 1.5
RM-β-CD	97.5 ± 0.6	94.9 ± 0.5	98.4 ± 1.2	100.3 ± 0.3

**Table 4 polymers-17-02850-t004:** Estimated binding and intermolecular energies of APX/CD ICs.

Inclusion Complex	Estimated Free Energy of Binding	Final Intermolecular Energy
DM-IC	−8.01 kcal/mol	−9.51 kcal/mol
HP-IC	−7.40 kcal/mol	−8.89 kcal/mol
RM-IC	−7.31 kcal/mol	−8.80 kcal/mol
SBE-IC	−7.22 kcal/mol	−8.71 kcal/mol

## Data Availability

The original contributions presented in this study are included in the article. Further inquiries can be directed to the corresponding author.

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
