# Peer review of "Spray-Dried Inclusion Complex of Apixaban with β-Cyclodextrin Derivatives: Characterization, Solubility, and Molecular Interaction Analysis"

_polymers, 2025, doi:10.3390/polym17212850_

Round 1
Reviewer 1 Report
Comments and Suggestions for Authors
The manuscript entitled “Spray-Dried Inclusion Complex of Apixaban with β-Cyclodex-trin Derivatives: Characterization, Solubility, and Molecular Interaction Analysis” addresses an important and timely topic. The experimental was well designed, and the conclusion is well supported by the findings. I recommend a minor revision of this paper.
1. The novelty of this study should be further highlighted in the Introduction section. Especially the unique contributions in terms of spray drying process optimization and molecular docking parameter selection should be further considered.
2.Molecular docking simulation is a highlight of this paper, but the authors need to further explain the rationality of the selected docking parameters and models, particularly whether the conformational changes of APX under different pH conditions are considered. In addition, it is suggested to combine experimental data to deeply discuss the correlation between the low binding energy of DM-β-CD and its dissolution effect, so as to enhance the scientificity and persuasiveness of the study.
3.For the molecular docking analysis, the rationality of the selected parameters should be further explained, and the correlation between low binding energy and dissolution effect should be discussed in combination with experimental data.
4.The conclusion should be improved, and more critical data results should be added to this section.
Reviewer 2 Report
Comments and Suggestions for Authors
Overall, this manuscript presents interesting data on beta-cyclodextrin derivative complexes with the poorly water-soluble drug apixaban using the spray-drying method. The experiments are well-designed, and the results and discussion are well-written.
Recommendations:
- Add arrows to the FTIR spectra to indicate the missing peaks of the drug in the inclusion complex.
- In Table 2, format the coefficient of determination (R²) with a superscript in the column header.
- Include the average particle size of the spray-dried inclusion complex for each formulation to enable comparison of the effects of beta-cyclodextrin type and drug loading on particle characteristics
Reviewer 3 Report
Comments and Suggestions for Authors
This manuscript presents a well-executed study investigating the use of spray-dried β-cyclodextrin inclusion complexes to enhance the solubility of the poorly water-soluble oral anticoagulant, Apixaban. The combination of phase solubility, comprehensive solid-state characterization, and molecular docking provides a robust mechanistic and practical foundation. The clear identification of DM-β-CD as the superior carrier, resulting in a ~78.7-fold increase in APX solubility in water, is a significant and valuable finding. However, several points should be addressed to further strengthen its rigor and clarity:
- While cyclodextrin-based solubilization is a well-established approach, the novelty of using DM-β-CD for apixaban should be emphasized more clearly in the Introduction. The authors should better highlight how their approach differs from or improves upon previous apixaban solubilization strategies.
- More details on spray-drying optimization would improve reproducibility. It would also help to provide justification for the chosen inlet/outlet temperatures, considering potential thermal degradation of apixaban.
- The study successfully demonstrates a dramatic increase in saturated solubility. However, for a poorly soluble drug intended for an oral dosage form, solubility enhancement is often less critical than the rate of dissolution. Since the major goal is to enhance APX bioavailability, the manuscript should include in vitro dissolution studies for the optimal ICs compared to pure APX and the physical mixture. This is essential to confirm the practical utility of the spray-dried complexes in a physiologically relevant context.
- The conclusion restates the findings well, but it would benefit from a short note on potential future applications or limitations.
- Figure legends: Ensure all abbreviations (e.g., PM, IC) are defined in each figure legend.
- Table 4: the title was missing.
